# Computer Aided Diagnosis of Melanoma Using Deep Neural Networks and Game Theory: Application on Dermoscopic Images of Skin Lesions

**DOI:** 10.3390/ijms232213838

**Published:** 2022-11-10

**Authors:** Arthur Cartel Foahom Gouabou, Jules Collenne, Jilliana Monnier, Rabah Iguernaissi, Jean-Luc Damoiseaux, Abdellatif Moudafi, Djamal Merad

**Affiliations:** 1LIS, CNRS, Aix Marseille University, 13288 Marseille, France; 2Research Cancer Centre of Marseille, Inserm, CNRS, Aix-Marseille University, 13273 Marseille, France; 3Dermatology and Skin Cancer Department, La Timone Hospital, AP-HM, Aix-Marseille University, 13385 Marseille, France

**Keywords:** melanoma detection, computer aided-diagnosis, convolutional neural networks, explainability, game theory, hierarchical architecture, XAI

## Abstract

Early detection of melanoma remains a daily challenge due to the increasing number of cases and the lack of dermatologists. Thus, AI-assisted diagnosis is considered as a possible solution for this issue. Despite the great advances brought by deep learning and especially convolutional neural networks (CNNs), computer-aided diagnosis (CAD) systems are still not used in clinical practice. This may be explained by the dermatologist’s fear of being misled by a false negative and the assimilation of CNNs to a “black box”, making their decision process difficult to understand by a non-expert. Decision theory, especially game theory, is a potential solution as it focuses on identifying the best decision option that maximizes the decision-maker’s expected utility. This study presents a new framework for automated melanoma diagnosis. Pursuing the goal of improving the performance of existing systems, our approach also attempts to bring more transparency in the decision process. The proposed framework includes a multi-class CNN and six binary CNNs assimilated to players. The players’ strategies is to first cluster the pigmented lesions (melanoma, nevus, and benign keratosis), using the introduced method of evaluating the confidence of the predictions, into confidence level (confident, medium, uncertain). Then, a subset of players has the strategy to refine the diagnosis for difficult lesions with medium and uncertain prediction. We used EfficientNetB5 as the backbone of our networks and evaluated our approach on the public ISIC dataset consisting of 8917 lesions: melanoma (1113), nevi (6705) and benign keratosis (1099). The proposed framework achieved an area under the receiver operating curve (AUROC) of 0.93 for melanoma, 0.96 for nevus and 0.97 for benign keratosis. Furthermore, our approach outperformed existing methods in this task, improving the balanced accuracy (BACC) of the best compared method from 77% to 86%. These results suggest that our framework provides an effective and explainable decision-making strategy. This approach could help dermatologists in their clinical practice for patients with atypical and difficult-to-diagnose pigmented lesions. We also believe that our system could serve as a didactic tool for less experienced dermatologists.

## 1. Introduction

Melanoma is still considered a serious public health issue. The only way of preventing its mortality is early diagnosis. In this field, prevention campaigns have shown a relatively interesting impact, but more screening actions must be carried out to increase early detection of melanoma and reduce its progression to an advanced stage. Dermatologists perform screening in their daily practice, but, unfortunately, medical resources are limited. Computer-aided diagnosis (CAD) systems are an interesting approach to assist dermatologists for melanoma screening in clinical practice [1].

Nowadays, with advances in deep learning, new perspectives of automating the process for the early detection of melanoma among pigmented lesions to assist dermatologists can be considered. Indeed, CAD based on CNNs have demonstrated high-quality performances, matching those of dermatologists in an experimental context [2,3,4]. Most of the common strategies proposed to improve the performance of skin cancer diagnosis can be grouped into three categories [5]: Transfer learning, data augmentation and ensemble learning. Transfer learning is used to improve learning by transferring knowledge from related tasks that have been learnt previously. In fact, the main reason behind the use of transfer learning for this context resides in the high similarity between malignant and benign lesions making their identification and classification very slow. Moreover, transfer learning is more effective in classifying similar lesions, making it a first choice [6]. The main way to apply transfer learning to skin lesions is to reuse the CNN’s architectures that have been pre-trained on the ImageNet dataset. Perez et al. [7] conducted an extensive experimental study in which they analyzed the effectiveness of three well-known optimization algorithms, as well as the performance impact of using transfer learning methods. Their  study confirmed the effectiveness of CNN combined with transfer learning for the melanoma diagnostic task. On the other hand, training a deep learning model requires a considerable amount of data, while the availability of annotated skin lesion images is often limited. Therefore, data augmentation is another well-used strategy to improve the model performance. In this context, Zhao et al. [8] used data augmentation to improve the accuracy of skin lesion classification. To this end, they used a data augmentation method based on generative adversarial networks to generate synthesized images for training a DenseNet201 model. More recently, Maron et al. [9] observed that increasing data can mitigate the effect of conflicting examples on the classification of skin lesions. Another successful technique to achieve high performance on skin lesions classification is to assemble a finite set of CNNs [10]. Mahbod et al. [11] developed and evaluated a multi-scale fusion technique based on the ensemble method. Their  approach used three CNN models trained on cropped images of different sizes and achieved 86.2% accuracy on the ISIC 2018 dataset. Foahom et al. [12] applied an ensemble method based on the directed acyclic graph technique for melanoma detection. Ensemble methods for building CAD for skin cancer detection were also used in [13,14,15,16,17].

Despite their performances, these CADs have not yet found their way into the clinical context due to a variety of reasons mentioned by Goyal et al. [18], among which we can mention the main ones. In fact, the generation of false negatives and the difficulty sometimes even for computer vision experts to understand the decisions made by deep learning frameworks accentuates the skepticism of end users, especially dermatologists. Therefore, in addition to improving the performance of current systems, the improvement of their explainability is another important challenge to achieve the goal of extending these systems to real clinical settings.

Regarding methods to improve the explainability of CADs, there are mainly three approaches in the literature: visualizing features maps, building content-based image retrieval (CBIR) systems and incorporating dermatologists’ knowledge. The core principle of visualizing features maps is to highlight the regions in an image that contribute to the CNN decision. The class activation maps [19,20] is the most popular method applied for this task. Van et al. [21] were the first to conduct such an analysis on skin cancers detection. They showed that CNNs were able to learn features similar to those used by dermatologists for diagnosis such as border and skin color. Works featured in [22,23] have also used Grad-CAM to provide explainability to their CAD systems. Another type of visualization is conducted using SHapley Additive exPlanations (SHAP) values [24], as in [25], to analyze predictions at a pixel-level and ensure that the model is looking at pertinent parts of the images. However, visualization approaches are revealed after the training of the model, which prevents any impact on the performance of the model. CBIR systems are an alternative approach to improve the explainability of CAD. The idea behind this approach is to improve the explainability of the decision making process by retrieving and displaying similar past cases relevant to the one being examined. Tschandl et al. [26] compare the diagnostic accuracy of a CBIR system based on dermatoscopic images to predictions made by a CNN. Their  results revealed that CBIR systems were able to outperform a CNN used alone. Allegretti et al. [27], in the same context, proposed to combine deep neural networks and embedding networks for dermoscopic image retrieval, which allowed them to obtain a better retrieval accuracy. CBIR systems for skin lesion detection are also used in [28,29]. However, such methods have two shortcomings [30]: first they suffered from the ’semantic gap’, meaning that feature similarity did not necessarily correlate to label similarity; secondly, they suffer from ’user-gap’, meaning that what a CNN considers as similar from a disease point-of-view does not necessarily correlate with human measures of similarity. The last approach to improve the explainability of CAD consists in modeling the knowledge and practices of dermatologists to integrate them into a CAD system. The works presented in [31,32] followed this approach. They structured their datasets following the taxonomic organization [33] of skin lesions to develop their framework. Nevertheless, despite the fact that this approach brings rationality to the decision, a recent study [34] explains that models trained using this approach performed worse than a simple multi-class model.

The goal of this study is to develop an accurate CAD for melanoma diagnosis while providing an explanation of its decision process. To this end, the proposed approach combines the three strategies to build an accurate CAD previously discussed. Indeed, the pipeline of this study is an ensemble method combining several models, so each of these models was developed from pre-trained architectures and was trained on data augmented with synthetic images obtained by artificial data generation techniques. Moreover, pursuing the other objective of this study which is to build an explainable CAD, our method introduces a new hierarchical framework inspired by game theory in order to build a decision process understandable by users and non specialists. In addition, not only has the framework been combined with a heatmap visualization allowing for a better interpretation of the results, but an innovative method to evaluate the confidence level of a prediction has also been introduced. This approach would allow our CAD to bring transparency in decision making, and improve its performance compared to previous methods.

## 2. Results

### 2.1. Results of Our Approach on the Test Dataset

We ran each of our experiments five times and the results were aggregated to generate an ROC curve of model performance. As shown in Figure 1, our framework performed well on all classes of lesions, far above a hazardous prediction (AUROC = 0.5). Our approach achieved a mean AUROC of 96% on the entire test dataset. Looking at the performance of each class individually, the class with the highest AUROC is the benign keratosis class with an AUROC value of 97%. The Nevus class obtained an AUROC of 96%. The class with the lowest performance was the melanoma class with an AUROC of 93%.

### 2.2. Comparing Our Approach with Prior Works

To evaluate our work, we compared the obtained performances with those obtained by prior works. For this, we selected a set of works developed within the same classification task, i.e., the multi-class classification of melanoma, nevus and benign keratosis. The reference works were also based in the use of CNNs. Table 1 presents the results of this study. Our approach outperformed all the compared methods by reaching an AUROC improvement of 5% for Melanoma, 4% for Benign Keratosis, and 8% for Nevi. Moreover, our approach led to a mean BACC of 0.86 which is 9% higher than the one obtained by the method proposed in [12].

### 2.3. Use Case and Performance Analysis

We further presented our results to a dermatologist collaborating with us in our laboratory for an evaluation in a real clinical context of our tool’s use case. We present in Figure 2 two illustrative examples chosen by the dermatologist. Lesion 1 (see (a) in Figure 2) is a dermoscopic image of a typical Melanoma with a high prediction ($p_{M}=1.00$) for this diagnosis and low prediction for the two other classes by our system (a probability of 0.00 for both Begnin keratosis and Nevi). The Heat-map presented a high activation on the area with white blue veil color, a typical region for Melanoma diagnosis. For this typical case, no other step was necessary, and the model seems efficient. On the other hand, lesion 2 represents a dermoscopic image of a misleading pigmented lesion. At first sight, this lesion could be a benign Keratosis or a melanoma. The first step indeed provided similar predictions for both Melanoma ($p_{M}=0.53$) and Benign Keratosis ($p_{B}=0.46$) and a lower prediction for Nevi ($p_{N}=0.01$) by s3 model with a confident score *u* of 54%. The second step of the framework classified more accurately between these two classes and sorted out a high prediction ($p_{M}=0.90$) for melanoma, which was the ground truth. In fact, the second heat-map focused on a slight regression area (the whiter part in the center of the region), a key region for the diagnosis. The second step trained specifically to distinguish Benign Keratosis and Melanoma was more efficient for this particular task and was able to correctly diagnose the melanoma lesion. These different stages of the framework highlight the ability of our approach to differentiate and classify difficult pigmented lesions as well as its ability to provide a more transparent decision process.

### 2.4. Performances of the Individual Model Used in Our Framework

Table 2 shows the results obtained with all the seven models for each individual task: MEL versus ALL, BEK versus ALL, NEV versus ALL, MEL versus NEV, MEL versus BEK, NEV versus BEK, and BEK versus MEL versus NEV. If we first look at the one-versus-all (ova) classifiers, we observe that the BEK-vs-ALL classifier is the one that has obtained the best performance with a BACC equal to 90%. This suggests that the class Benign Keratosis is the one that best distinguishes itself from the other two classes Nevi and Melanoma. On the other hand, with the one-versus-one (ovo) classifiers, the NEV-vs-BEK classifier performed best with a BACC of 94%. Nevus and keratosis are the two easiest classes to discriminate in our task. The most difficult lesions for our framework to distinguish are Nevi from Melanoma where the classifier has obtained a BACC of 87%. The 3-class model reached a BACC of 84%, which achieved a 2% improvement with the entire pipeline.

### 2.5. Ablation Study: Choice of the Best Hyper-Parameters $u_{1}$ and $u_{2}$

We present in Table 3 the results of the grid-search we have done on the validation set to find the best values of the hyper-parameters to maximize the detection of Melanoma. For this analysis, we used only the models obtained in the run that achieved the best performance. The best combination value were obtained for $u_{1}$ and $u_{2}$ having respectively the value 0.1 and 0.5.

## 3. Discussion

In this study, a novel deep learning ensemble method is presented to obtain an accurate and explainable CAD of melanoma. A framework following a hierarchical structure and combining seven CNNs has been created for this purpose. The BACC and AUROC scores were mainly used to analyze classification performance. The proposed method reached an average AUROC of 96%, which demonstrates the good performance of our approach in this task. The performance of our approach was also compared to that of previous works in the same task and outperformed all of them with a minimum margin of 9% in terms of BACC. A use case analysis is also performed by a dermatologist to assess the decision-making transparency of our approach in a clinical setting.

In our study, we integrate all successful strategies to obtain an accurate classification of skin lesions. Similar to the works of [10,38,39], we used the pre-trained EfficienetNet architecture as the backbone of all our models. We also applied data augmentation as in [8,9] to increase the robustness of our models. Finally, we developed a pipeline that combines several models to build an ensemble learning which is a well-known strategy used in previous works [9,12,15,16].

Our approach integrates the heat-maps visualization in the framework. This heat-maps visualization showing the arrangement between the feature maps and the visual input is a particularly popular way of explaining CNNs [40]. In this context, Van et al. [21] has shown that the Grad-CAM method reveals features similar to those used by dermatologists to make their diagnosis. Moreover, the definition, for the first time, to our knowledge, of a method to evaluate the confidence-level of a prediction brings more clarity to the prediction made by a CAD and provide dermatologists with enough elements to help them make the best diagnosis. Indeed, as indicated in [40], combining different methods of explanation with various modalities makes it possible to obtain more complete explanations. Furthermore, providing clinical end users with sufficient evidence to support the prediction builds confidence in the model’s decision and helps dermatologists identify potentially questionable decisions [41].

This study responds to the requirement of CADs that are explainable since this criteria become more relevant considering the recent ethical and legal standards [42,43]. Nevertheless, our study has some limitations that need to be addressed. First, in this study, we slightly approached the resolution of the class imbalance problem [44] which is present in computer vision task and more particularly in skin lesions classification. Future work integrating tricks for optimizing the performance of deep models about this aspect, such as those presented in [10,45], could strengthen the robustness of our method. Second, our study lack of enough investigation on the usability and adoptability of our application in real clinical scenario as suggested in [46]. Even though we have integrated an analysis of cases of use of our system by a dermatologist, it would be wise to conduct a real clinical study by integrating a large number of clinicians.

The results obtained in this study show that it is clearly possible to build automatic diagnostic systems based on deep learning that are explainable without losing diagnostic accuracy. We have made the codes and models used in this manuscript available online to the community. They are accessible via the link provided in the Appendix A.

## 4. Methods and Materials

In this section, we present our methodology. We describe the overall workflow of our framework and the CNN backbone model used in our experiments.

### 4.1. Convolutional Neural Network

Several CNN architectures are open-source, with some of them being already trained on the ImageNet dataset. Thus, we can reuse their weights and biases and fine-tune them for application in our task. This is known as transfer learning. EfficientNet networks [47] are currently one of the most commonly used architectures in computer vision. Their  authors defined a way to scale the models when more computing power is available. They proposed for that eight versions of the architecture depending on the scale level ranging from B0 to B7. The B7 version achieved 84.4% top-1 accuracy on ImageNet while being 8.4× smaller and 6.1× faster on inference than the best existing CNN at that time. They have also been successfully used in the task of classifying skin lesions [13,48]. In our works, we use the B5 version of EfficientNet as the backbone of our models due to the available resources. We modified the original model by replacing the classification layer with a new fully connected (FC) layer of two nodes to perform binary classification or three nodes to perform ternary classification. The news layers were initialized with a Kaiming initialization [49].

### 4.2. Game Theory

Game theory is a theoretical framework for modeling conflict situations among competing players and for analyzing the behavior of various players. Game theory was originally developed as a mathematical model in the field of operations research and has been applied to other disciplines to solve competition and collaboration problems between different objectives. More recently, some researchers investigated the use of game theory for deep learning [50]. There are three main components in the game: the players, the strategies they use, and the payoff they receive from the corresponding strategies [51]. In general, the principle of game theory can be summarized as the process by which the decision-maker makes a choice to maximize the benefit of each player after the opponent adopted a certain strategy, assuming that all players are rational. In the process of the game theory in this study, the CNN models represent the players, two strategies are defined for the players, the first is to cluster the predictions by confidence level (confident, medium, uncertain) and the second is to make the final prediction. The payoff value depends on the output probabilities of the models.

### 4.3. Class Activation Map

We adopted the Gradient-weighted Class Activation Mapping (Grad-CAM) [20] algorithm to visualize the region contributing to the CNN model’s decisions as a heat map. Let the penultimate CNN layer produce *K* feature maps $A^{k}\in R^{u*v}$ of width *u* and height *v*. We denote by $y^{c}$ the score produced by the feature maps $A^{k}$ spatially pooled using a global average pooling GAP. Grad-CAM algorithm generated heat-map using the following formula:(1)$$L_{Grad-CAM}^{c}=ReLU\left(\sum_{k}w_{k}^{c}A^{k}\right)$$

$w_{k}^{c}$ represents the gradients of $y^{c}$ with respect to feature maps *A*, while ReLU is the rectified linear unit function. In this study, we used the last convolutional layer to compute the weights as suggested by [20].

### 4.4. Description of Our Framework

We designed a novel framework (see Figure 3) following a hierarchical architecture from the combination of several CNNs. We combined seven CNNs among which one was trained in a multi-class task to classify Melanoma among Nevi and Benign Keratosis. Three of the CNNs were trained to recognize a given diagnosis class of lesions with an ova strategy. The three others are trained to distinguish two classes of lesions with an ovo strategy. The CNNs trained with an ova strategy are used only to evaluate the confidence level of the multi-class CNN and generate a confident score.

For a sample image *x*, we consider a CNN to be a function $s:x\Rightarrow R^{n}$ that generates a vector p of size *n* containing predicted probabilities $p_{i}$ that *x* belongs to the class *i*, where *n* represents the number of classes, $p_{i}\in \left[0,1\right]$ and $\sum_{n}p_{i}=1$. The classifier *s* makes the decision to classify *x* in class *i* based on Equation (Equation 2).
(2)$$class\left(x\right)=i,\;if\;max\left(p\right)=p_{i}.$$

We used the notations $s_{3}$, $s_{i\bar{i}}$, and $s_{ij}$, respectively, for the 3-class CNN, the CNN trained with the ova strategy and the CNN trained with the ovo strategy.

$s_{3}$ generates three predicted probabilities $p_{B}$, $p_{M}$ and $p_{N}$ that *x* belongs to classes *B*, *M* and *N*. The classes indicate the nature of the lesion, *B* for Benign Keratosis, *M* for Melanoma and *N* for Nevi. Each $s_{i\bar{i}}$ is specialized in one class *i* and generates two predicted probabilities $p_{i}^{{}^{\prime }}$ and $p_{\bar{i}}^{{}^{\prime }}$ that *x* respectively belongs to class *i* or not. $s_{ij}$ generates two predicted probabilities $p_{i}$ and $p_{j}$ that *x* respectively belong to class *i* or *j*.

To evaluate the confidence level of a prediction $p_{i}$, we introduce the function *u* defined by:(3)$$u\left(p_{i}\right)=abs\left(p_{i}-p_{i}^{{}^{\prime }}\right)$$

The function *u* estimated the level of confidence of prediction by calculating the absolute error of the predictions made by $s_{3}$ and $s_{i\bar{i}}$.

In the first step, $s_{3}$ and the three set of $s_{i\bar{i}}$ are used to cluster each input image *x* into three groups: high confidence, medium confidence and uncertain predictions. The division by group is based on Equation (Equation 4). $\alpha_{1}$ and $\alpha_{2}$ were determined by grid-search according to the balanced accuracy obtained by the framework on the validation set.
(4)$$group\left(x\right)=\left\{\begin{array}{cc} g1:high & for\;u\left(p_{i}\right)\leq \alpha_{1}\\ g2:medium & for\;\alpha_{1}\leq u\left(p_{i}\right)\leq \alpha_{2}.\\ g3:uncertain & \mathrm{otherwise}. \end{array}\right.$$

When the prediction belongs to g1, the prediction of $s_{3}$ is kept and no further steps are performed. On the other hand, if the prediction belongs to g2, the classifier $s_{ij}$ is used to predict the class of *x*. For uncertain cases, the three classifiers $s_{BM}$, $s_{BN}$ and $s_{MN}$ are used to predict the class of *x* based on the Max-Win rule [52]. Algorithm 1 describes the steps followed by our framework to predict the class of an observation *x*.
**Algorithm 1** Pseudo-code of our framework.**Require:** Image *x*, 3 pairwise CNNs $s_{ij}$, 1 3-class CNN $s_{3}$, 3 pairwise CNNs $s_{i\bar{i}}$, list of the three classes class_list=[1,2,3]
**Ensure:** predicted class of *x*
 Generate the prediction probability $p_{i}$ of the class associated to *x* with $s_{3}$
 Generate the probability $p_{i}^{{}^{\prime }}$ of belonging to the class *i* by the model $s_{i\bar{i}}$ specialized to this class
 Estimate the confidence level $u(p_{i})$ of the prediction made by $s_{3}$ using the equation $u\left(p_{i}\right)=abs\left(p_{i}-p_{i}^{{}^{\prime }}\right)$
 **if**
$u\left(p_{i}\right)<\alpha_{1}$
**then**
  categorizes the prediction as being certain
  *x* belongs to class *i*
 **else if**
$\left(\alpha_{1}<u\left(p_{i}\right)<\alpha_{2}\right)$ & $\left(p_{i}>p_{j}>p_{k}\right)$
**then**
  Categorizes the prediction as medium confident
  Generate prediction probabilities $p_{i}$ and $p_{j}$ made by the model $s_{ij}$
  **if**
$p_{i}>p_{j}$
**then**
   *x* belongs to class *i*
  **else**
   *x* belongs to class *J*
  **end if**
 **else**  Categorizes the prediction as being uncertain
  Generate prediction probabilities of all the 3 pairwise CNNs $s_{ij}$
  *x* belongs to class obtain by applying Max-Win rule on the 3 pairwise CNNs $s_{ij}$
 **end if**


Theses predictions are normalized using the Equation (Equation 5) to assign final prediction probability.
(5)$$prob\left(i\right)=\frac{p_{i}}{p_{i}+p_{j}+p_{k}};\;i,j,k=B,M,N\;and\;i\neq j\neq k$$

### 4.5. Experimental Setup

#### 4.5.1. Dataset Preparation and Preprocessing

We performed this work on the publicly available dataset ISIC 2018 [53,54]. ISIC 2018 contains 10 017 images annotated into seven classes: Actinic keratosis, Basal Cell Carcinoma, Melanoma, Benign keratosis, Dermatofibroma, Nevi and Vascular lesion. We focused our work on the three most challenging classes of lesions for the detection of Melanoma. This resulted in a dataset of 8 917 lesions: Melanoma (1 113), Nevi (6 705) and Benign Keratosis (1 099). We started by randomly dividing the dataset into 70% training images, 10% validation images, and 20% test images. Then, to reduce the effect of imbalance between different classes, we used a data augmentation strategy based on horizontal flip, vertical flip, rotation, width and height shift. The distribution of the dataset is shown in Table 4.

Once the dataset is loaded, a preprocessing is engaged to reduce the effects induced by the different acquisition setups such as the lighting difference and the presence of artifacts. We applied standard preprocessing techniques for skin lesion images [12]. We resized our images to 456 × 456 using a bicubic interpolation, and performed color standardization using the gray world color constancy algorithm [55] to deal with color variability.

#### 4.5.2. Fine-Tuning the Networks

We fine-tuned only 64 percent of the deeper layers of our pretrained models with a batch size of 32 during 150 epochs. We used Adam optimizer to update the weights and biases of our networks at every iteration to minimize the loss function output. The general Adam term employed in our work is defined as [56]:(6)$$\theta_{t}\leftarrow \theta_{t-1}-\alpha .m_{t}/\left(\sqrt{v_{t}}+\hat{\epsilon }\right)$$
where θ is the parameter vector of the network, *t* represents the iteration number, α is the learning rate, and $m_{t}$ the momentum term.

We calculated the loss value using a weighted cross entropy function. The general term of the cross-entropy loss is:(7)$$L=-w_{i}\sum_{n=1}^{N}p.log\left(q\right)$$
where *p* is the ground-truth label, *q* is the predicted Softmax probability, $w_{i}$ is the weight for class *i* and *N* is the number of classes. $w_{i}$ corresponds to the inverse number of samples for each class as proposed in [57]. Similar to [58], we used the cyclical learning rate (CLR) proposed by [59] to schedule the learning rate during training in the range from 0.001 to 0.00001. We opted for the “triangular2” setting of CLR with a step size of 2000 iterations. During fine-tuning, we also applied regularization to avoid overfitting by stopping the training early when the BACC on the validation set did not improve after 15 epochs.

We performed experiments using the Pytorch software library (1.11.0) under the Ubuntu 18.04 operating system. Specifically, all experiments were performed on a workstation with an Intel(R) Core CPU (3.20 GHz) and one graphic card. The graphics card is an Nvidia GeForce GTX 2080 GPU with 8 GB of memory. The time required to train and validate each of the seven models on this machine varied from 10 to 15 h. The time required to test the entire pipeline was approximately one minute.

#### 4.5.3. Metrics

We used the AUROC score to evaluate the performance of our framework. The dataset presents a skewed distribution where a normal accuracy would favor the correct classification of the over-represented class Nevi. Thus, we also opted for the BACC metric as another measurement for our experiments. The BACC is defined as:(8)$$\mathrm{BACC}=\frac{\sum_{i}Sensitivity_{i}}{N}$$

In Equation (Equation 8), *N* represents the total number of classes in the task, and $Sensitivity_{i}$ represents the sensitivity of class *i*.

## 5. Conclusions and Future Works

Nowadays, one of the main goals of CAD in the medical field is not only to achieve high performance, but also to improve the explainability of these systems and increase their use in the clinical setting. In this study, we present a new hierarchical ensemble deep learning framework for melanoma detection from dermoscopic images, aiming to both improve the performance of the existing systems and provide more clarity in its decision process. The proposed approach combines seven CNN models through game theory and combines it with heatmap visualization. We also introduced a new method to evaluate the confidence level of a prediction generated by an automated system and integrated it into our framework. The results show that our approach can effectively improve the accuracy of CAD compared to the state of the art. Furthermore, by conducting a use case study of our framework by a dermatologist, we could observe that the decision process of our approach was found to be more intuitive and explainable, which would support its use in real clinical settings. Future work will focus on two main directions. The first direction is to improve the predictive performance of our framework, with a particular focus on reducing false negative rates and mitigating class imbalance issue. The second direction will be to perform a large-scale clinical validation of our approach. At this stage, our system has great potential for use in real clinical settings as a training tool for novice dermatologists.

## Figures and Tables

**Figure 1 ijms-23-13838-f001:**
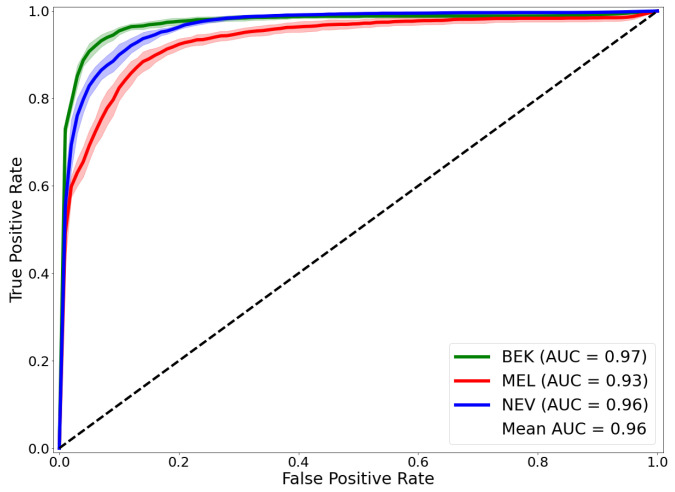
Receiver operating characteristic (ROC) curves obtained with our framework. The Area under the curve (AUC) of the ROC is given for each lesion class: Melanoma (MEL), Benign Keratosis (BEK) and nevi (NEV) on the test set.

**Figure 2 ijms-23-13838-f002:**
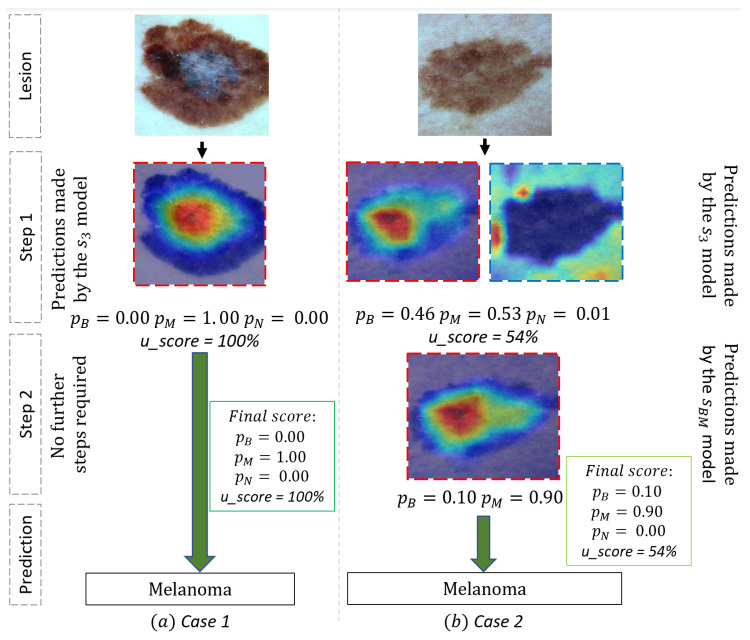
Use cases of our CAD system for classification of Melanoma (M), Nevi (N) and benign Keratosis (B) illustrated by two examples. The dermoscopic image of Lesion 1 represents a typical Melanoma that was associated with a high prediction (1.00) to a Melanoma at the first stage of our framework with a confidence score of 100%. Indeed, the heat-map in the first step focuses on the regression area which is a typical area for Melanoma diagnosis. On the other hand, the framework is very interesting for difficult Melanoma. The dermoscopic image of Lesion 2 is a suspicious lesion, but is not a typical melanoma and could be confused with benign Keratosis. The first step presented a shared prediction between benign Keratosis (0.46) and Melanoma (0.53) with a confidence score of 54%. Being in a situation where the confidence level is medium, our framework directed the prediction of the Lesion 2 to the binary classifier trained to dissociate the two most probable classes in occurrence the benign keratosis versus melanoma ($s_{BM}$). The second stage allowed to refine the prediction by finally associating Lesion 2 to Melanoma with a much better probability (0.90). Indeed, the second heat-map focused on a key region for Melanoma. Heat-map generation is implemented with Grad-CAM [20]. Heat-map images framed by red dashed lines are those representative of the melanoma class; those framed by blue dashed lines are those representative of the benign keratosis class.

**Figure 3 ijms-23-13838-f003:**
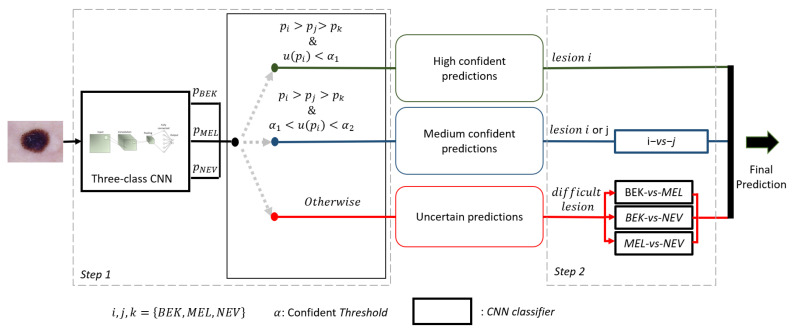
Our novel framework to ternarily classify Melanoma (MEL), Nevi (NEV) and Benign Keratosis (BEK). An image of a skin lesion is spent in a first set consisting of three CNNs trained to classify one lesion versus the two other lesions. The second step, depending on the group in which the image was previously placed based on the confidence score, provides the final decision with the associated probability.

**Table 1 ijms-23-13838-t001:** Comparison with existing CAD system for the multi-class classification of Melanoma, Nevi and Keratosis.

Works	BACC	Mean-AUROC	MEL-AUROC	BEK-AUROC	NEV-AUROC
Harangi et al. [35]	-	0.85	0.84	0.87	0.84
Bisla et al. [36]	-	0.92	0.88	-	-
Barata et al. (2019) [37]	0.70	0.88	-	-	-
Barata et al. (2021) [32]	0.74	0.92	0.80	0.92	0.85
Foahom et al. [12]	0.77 ± 0.00	-	0.87	0.93	0.88
Proposed framework	0.86 ± 0.01	0.96 ± 0.00	0.93 ± 0.01	0.97 ± 0.01	0.96 ± 0.00

**Table 2 ijms-23-13838-t002:** Performance on the test set for each of the seven models on different tasks involved in our framework. We report on table the BACC and the AUROC scores.

Task	BACC	AUROC
BEK vs. MEL vs. NEV	0.84 ± 0.01	0.96 ± 0.00
MEL vs. ALL	0.81 ± 0.02	0.94 ± 0.01
NEV vs. ALL	0.86 ± 0.01	0.96 ± 0.00
BEK vs. ALL	0.90 ± 0.01	0.98 ± 0.00
MEL vs. NEV	0.87 ± 0.01	0.95 ± 0.01
MEL vs. BEK	0.91 ± 0.01	0.97 ± 0.00
NEV vs. BEK	0.94 ± 0.01	0.99 ± 0.00

**Table 3 ijms-23-13838-t003:** Result of the best combination of hyper-parameters $\alpha_{1}$ and $\alpha_{2}$ obtained with grid-search on the validation set.

($\alpha_{1}$, $\alpha_{2}$)	MEL-AUROC
(0.3, 0.5)	0.95
(0.3, 0.4)	0.95
(0.2, 0.5)	0.95
(0.2, 0.4)	0.95
(0.2, 0.3)	0.95
(0.1, 0.5)	0.96
(0.1, 0.4)	0.95

**Table 4 ijms-23-13838-t004:** Distribution of the dataset.

	Benign Keratosis	Melanoma	Nevi
ISIC 2018	1099	1113	6705
Ratio	0.12	0.12	0.75
Training set	769	779	4694
Generated data from training set	1231	1221	306
Final training set with data generated	2000	2000	5000
Validation set	110	111	670
Test set	220	223	1341

## Data Availability

The dataset is available online at ISIC 2018 (https://challenge2018.isic-archive.com/task3/training/, accessed on 8 November 2022).

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
