# Peer review of "Computer Aided Diagnosis of Melanoma Using Deep Neural Networks and Game Theory: Application on Dermoscopic Images of Skin Lesions"

_ijms, 2022, doi:10.3390/ijms232213838_

Round 1

Reviewer 1 Report

This study is about finetuning existing artificial intelligent models for computer aided diagnosis of melanoma based on publicly available datasets. As noted in the Introduction, many similar works have been available elsewhere but there is a lack of clinical evaluation of the models. For this study, the authors should cover the clinical evaluation of their finetuned model. However, again, the authors did not address the stated knowledge gap. Also, no discussion section exists in this paper. All in all, I cannot see any value about this study despite modifying several layers of the other researchers’ models and having it tested with publicly available datasets. This study cannot advance the knowledge of the topic area at all. Unless the finetuned model is tested in clinical settings, it needs to be rejected. Please see my detailed comments below.

1.    Abbreviations, e.g. CAD, CNNs, etc. need to be defined before using. Please review this issue for the whole manuscript.
2.    The in-text citations are not in proper order, e.g. [33,34] noted in Figure 1, etc. Please review this issue for the whole manuscript.
3.    Methods and materials should be presented before the Results.
4.    Lines 302-303: Please provide details of the central processing and graphical processing units.
5.    Methods and materials: The time required for training, validating and testing the model should be reported.
6.    Methods and materials: As noted in the Introduction, lots of studies have investigated this topic area but there is a lack of clinical evaluation. This work just finetuned existing models and evaluated the finetuned model with publicly available datasets. Hence, its value appears limited. Also, no dermatologist was involved for the model evaluation. From my own perspective, this study does not provide any substantial knowledge contribution at all.
7.    Discussion: No such section exists in this paper which is another major issue.

Reviewer 2 Report

The authors presented Computer Aided Diagnosis of Melanoma using Deep Neural Networks and Game Theory.

The topic is interesting & problem is truly identified but the possible solution has several issues.

1. First the sequence of article is not standard.

2. The motivation of the paper needs to be explained in the literature review

3. Figures labels are not comprehensive. The amount of information presented in these labels does not detail what the figures intend to present.

4. What are major contributions, please write specifically.

5.  Results are presented before methodology !

6. Please include pseudo code of your methodology for easy understanding of readers.

7. Analysis of results section is weak.

8. No comparisons in state of art is reported.

Reviewer 3 Report

1. Overview and general recommendation:

The research problem was formulated correctly, and the researcher's methodology was also precisely presented. I have no comments on the scientific part.

 The presented material corresponds to the profile of the "International Journal of Molecular Sciences". The scientific value of the submitted material qualifies the article for publication in this Journal.

 The article may be published after completing and correcting all issues. I recommend that a major revision is necessary. I made the detailed comments in point 2.  I ask that the authors specifically address each of my comments in their response.

 2. Major comments

 The layout of the manuscript is incorrect. In particular, there are no Results and discussion.

 Please consider changing the layout of the manuscript by introducing the main points:

1.     Introduction

2.     Materials and methods

3.     Results and discussion

4.     Conclusions

 Conclusions must be formulated precisely and correspond with the obtained research results. In particular, they should display elements of novelty in the context of using the results of research and analysis. In this situation, the article presents an interesting research methodology, which, however, does not bring revolutionary elements in the context of innovative methods and technologies in scientific research in the subject area of the undertaken research problem. Therefore, it is recommended to clearly indicate the novelty of the proposed solution.

 To sum up, the "Discussion" and "Conclusions" should be formulated in such a way as to present the key results obtained in effect of the completed research using proprietary methods.

Round 2

Reviewer 1 Report

Improvements are noted in this revised manuscript. However, the following comments should be addressed prior to publication.

1.    Lines 60-68 should be moved to the Conclusion section.
2.    Lines 69-73 should be removed.
3.    Writing style should be improved. For example, it is unnecessary to repeat the point, "Inspired from/by game theory" multiple times. Informal expressions should be avoided, e.g. "Thanks to...", etc. No reference should be cited in the Conclusion part as no new idea should be introduced there. These are just examples. Please review the whole manuscript and make any necessary changes.
4.    Discussion: You should compare your "innovative" approach with the generative adversarial network (GAN) framework as both are based on game theory and GAN can be used for computer aided diagnosis as well.
5.    Lines 387-407: References should be cited for these paragraphs.

Reviewer 2 Report

Accepted in present form.

Reviewer 3 Report

I have no comments. The scientific value of the submitted material qualifies the article for publication in this form. 
